# APP Maturation and Intracellular Localization Are Controlled by a Specific Inhibitor of 37/67 kDa Laminin-1 Receptor in Neuronal Cells

**DOI:** 10.3390/ijms21051738

**Published:** 2020-03-04

**Authors:** Antaripa Bhattacharya, Adriana Limone, Filomena Napolitano, Carmen Cerchia, Silvia Parisi, Giuseppina Minopoli, Nunzia Montuori, Antonio Lavecchia, Daniela Sarnataro

**Affiliations:** 1Department of Molecular Medicine and Medical Biotechnology, University of Naples “Federico II”, Via S. Pansini 5, 80131 Naples, Italy; antaripa1210@gmail.com (A.B.); ad.limone@studenti.unina.it (A.L.); silvia.parisi@unina.it (S.P.); giuseppina.minopoli@unina.it (G.M.); 2Department of Translational Medical Sciences, University of Naples “Federico II”, Via S. Pansini 5, 80131 Naples, Italy; napolitanofilomena07@gmail.com (F.N.); nmontuor@unina.it (N.M.); 3Department of Pharmacy, “Drug Discovery Lab”, University of Naples “Federico II”, Via D. Montesano 49, 80131 Naples, Italy; carmen.cerchia@unina.it (C.C.); antonio.lavecchia@unina.it (A.L.)

**Keywords:** Amyloid precursor protein (APP), intracellular trafficking, 37/67 kDa laminin receptor, prion, posttranslational modifications, small molecule inhibitors

## Abstract

Amyloid precursor protein (APP) is processed along both the nonamyloidogenic pathway preventing amyloid beta peptide (Aβ) production and the amyloidogenic pathway, generating Aβ, whose accumulation characterizes Alzheimer’s disease. Items of evidence report that the intracellular trafficking plays a key role in the generation of Aβ and that the 37/67 kDa LR (laminin receptor), acting as a receptor for Aβ, may mediate Aβ-pathogenicity. Moreover, findings indicating interaction between the receptor and the key enzymes involved in the amyloidogenic pathway suggest a strong link between 37/67 kDa LR and APP processing. We show herein that the specific 37/67 kDa LR inhibitor, NSC48478, is able to reversibly affect the maturation of APP in a pH-dependent manner, resulting in the partial accumulation of the immature APP isoforms (unglycosylated/acetylated forms) in the endoplasmic reticulum (ER) and in transferrin-positive recycling endosomes, indicating alteration of the APP intracellular trafficking. These effects reveal NSC48478 inhibitor as a novel small molecule to be tested in disease conditions, mediated by the 37/67 kDa LR and accompanied by inactivation of ERK1/2 (extracellular signal-regulated kinases) signalling and activation of Akt (serine/threonine protein kinase) with consequent inhibition of GSK3β.

## 1. Introduction

The accumulation of amyloid beta peptide (Aβ) in the brain is a neuropathological hallmark of Alzheimer’s disease (AD) [1]. Aβ is produced by a sequential processing of amyloid precursor protein (APP) by *β*- and *γ*-secretases [2,3]. A large body of evidence suggests that the production of Aβ occurs in the endo/lysosomal pathway where APP physiologically traffics [4] and where *γ*-secretases are preferentially distributed [5,6]. Interestingly, deacidification of endosomal/lysosomal system decreases Aβ production [7]. Beside the previously described role of APP endocytosis in Aβ production [8,9], many reports suggest a critical role for intracellular trafficking of APP in Aβ generation [10,11,12,13,14,15,16,17] and indicate the trans-Golgi network as the main sorting station of APP to lysosomes, where the Aβ can be produced [18]. An aberrant production of Aβ in AD brains could amplify its neurotoxic effects through activation of GSK3β (Glycogen synthase kinase 3 beta) [19]. In fact, the neurotoxic Aβ peptide has been shown to activate GSK3β in hippocampal neurons and to cause cell death [20]—an event that is blocked by introduction of antisense oligonucleotides to GSK3β [21].

Previous reports revealed that the 37/67 kDa laminin receptor (LR) is involved in APP processing through a direct interaction with the *γ*-secretase complex (specifically with Presenilin-1, PS1) and indirect with *β*-secretase BACE1 [22].

Interestingly, knockdown of 37/67 kDa LR using anti-LR shRNA or anti-LR antibody (IgG1iS18) resulted in a significant reduction of Aβ levels, Aβ uptake and Aβ-induced cytotoxicity [23,24,25], suggesting 37/67 kDa LR is a novel target for AD [26].

Starting from these observations, we decided to test the effects of a specific 37/67 kDa LR inhibitor, NSC48478, in mouse GT1 and human SHSY5Y neuronal cell lines. We found that this compound, previously shown to be a strong inhibitor of 37/67 kDa LR binding to laiminin-1 [27], was able to hamper both the maturation of APP and the interaction of the receptor with APP, as well as APP intracellular distribution. Under inhibitor treatment, we found a partial retention of APP in the endoplasmic reticulum (ER), where acetylated APP immature isoforms accumulate. This phenotype was accompanied by APP redistribution into recycling Transferrin (Trf)-positive endosomes. Inhibition of vesicular acidification by NH_4_Cl completely rescued the effects exerted by the inhibitor on APP maturation, relocating APP into the “physiological” endo-lysosomal pathway. Moreover, by short hairpin RNA silencing of 37/67 kDa LR, we show that the effects of the small molecule inhibitor were mediated by the receptor, whose interaction with APP was significantly reduced by the small organic compound.

In addition, NSC48478 induced inactivation of ERK1/2 signalling and phosphorylation of Akt with consequent inactivation of GSK3β.

Altogether, these results reveal NSC48478 may act via a pH-sensitive compartment, through which APP normally traffics [4] and may be tested in disease conditions to test Aβ production and the consequent signal transduction events.

## 2. Results 

### 2.1. Maturation of APP Is Affected by 37/67kDa LR Inhibitor NSC48478

During its transit from the ER to the Golgi apparatus, newly synthesized APP can undergo a series of posttranslational modifications, such as *N*- and *O*-linked glycosylation, phosphorylation, sulfation [14,28] and acetylation [29].

In order to evaluate the possible effects of NSC48478 on APP levels, as well as on APP posttranslational modifications, we employed the neuronal GT1 cells we had previously used to investigate prion protein and 37/67 kDa LR trafficking under the use of receptor inhibitors [30]. 

APP migrated on SDS-PAGE gel as a typical glycosylated protein, showing different bands ranging from ~110 to ~135 kDa (Figure 1A), as already reported for other cell types [28].

As previously observed in the case of receptor cell adhesion to laminin-1, where the calculated IC_50_ value was 19.35 μM [27], we found that, without changing the total amount of protein, the compound exerted a strong effect on the maturation of APP at 20 μM concentration, with accumulation of the immature APP, which has a faster electrophoretic mobility respect to *N*- and *O*-glycosylated isoforms (mAPP) [31]. 

This phenomenon in GT1 cells seemed to be specific for APP, since it has been observed to a lesser extent or not at all, for other glycosylated proteins such as the prion protein PrP (Appendix A, lanes 1–2, PrP) and the placental alkaline phosphatase PLAP, respectively (Appendix A, lanes 1–2, PLAP). Moreover, the inhibitor did not exert any effects on APP maturation in another neuronal cell line, such as SHSY5Y cells (Appendix A, panel A).

Therefore, to evaluate whether in GT1 cells accumulation of immature APP by NSC48478 treatment in fact corresponded to immaturely glycosylated APP, different glycosidases were employed, followed by Western blot analysis with anti-APP antibody. 

Endoglycosidase-H (Endo-H) cleaves *N*-linked mannose-rich oligosaccharides in immature glycoproteins that traffic through the early secretory pathway between the ER and the first cisternae of the Golgi complex, where in the late cisternae, highly processed complex oligosaccharides are added to glycoproteins and cannot be digested by Endo-H. 

We found that incubation with this enzyme resulted in APP largely unaffected, both in the cell extracts from control (untreated cells) and treated with NSC48478 (Figure 1B, Endo-H, lanes +). Based on this result and on the fact that APP has been demonstrated to be *N*-linked complex glycosylated in other cell types [31,32], we decided to analyse the digestion with PNGaseF, which hydrolyzes nearly all types of *N*-linked sugar chains from glycoproteins, unless there was an α(1–3) Fucose on the core GlcNAc of the protein [33]. 

While on SHSY5Y cell extracts, the PNGaseF exerted the expected effect producing the loss of mature APP band (Appendix A, panel A), as well as the typically glycosylated prion protein PrP which resulted sensitive to PNGaseF treatment [34,35] (Appendix A, PrP), we found that in GT1 cells, APP was again largely unaffected both in untreated cells (without NSC48478) and in inhibitor treated cells (Figure 1B, PNGaseF), suggesting two possibilities: (1) the endogenous APP produced in this type of cells is mainly *O*-glycosylated, or (2) PNGaseF cannot cleave because an α(1–3) Fucose is on the core GlcNAc [33]. 

It has been reported that APP is a substrate for *O*-glycosylation modifications [36]. Since *O*-glycosidase removes *O*-linked galactose-*N*-acetylgalactosamine disaccharides after cleavage of the terminal sialic acids by Neuraminidase [37], we performed deglycosylation assays by using *O*-glycosidase and Neuraminidase. As expected, the use of *O*-glycosidase alone did not produce any effect on APP migration on SDS-PAGE (Figure 1B, compare lane 1 with 2). Incubation with neuraminidase produced a clear shift in the electrophoretic mobility of APP, especially when coincubated with *O*-glycosydase (Figure 1B, lanes 3 and 4). In contrast, immature APP accumulated under NSC48478 (bands indicated with #, lane 5) seemed unaffected by digestion with these two enzymes (Figure 1B, lanes 6 and 7 compared to 5). Conceivably, the band of immature APP (asterisk) is the unglycosylated form, while the intermediate bands (indicated with **#**) are not *O*-glycosylated APP isoforms. 

To understand why APP resulted resistant to PNGaseF digestion and whether the mature APP (mAPP) and the intermediate bands (#) derive from *N*-glycosylation events, we employed tunicamycin (Appendix A), a molecule known to prevent *N*-glycosylation events. According to previous observations [38], treatment of GT1 cells with tunicamycin for 16 h (Appendix A) revealed that the mature APP (mAPP) results from *N*-glycosylation modifications (and *O*-glycosylation, as shown in Figure 1B) and that after NSC48478 treatment (Appendix A), the resulting immature APP isoforms (* and #) were not *N*-glycosylated. The typically *N*-glycosylated PrP was carried as control of the procedure. 

Sensitivity of APP to tunicamycin but not to PNGaseF suggests that APP can be differently glycosylated in GT1 cells respect to other cell lines (such as SHSY5Y cells), where APP sugar modifications can be digested by PNGaseF (see Appendix A). Further experiments will be needed to validate this hypothesis.

Conceivably, the band of immature APP (*) is the unglycosylated form and the upper bands (indicated with #) could be attributed to other post-translational modifications, presumably phosphorylation or acetylation that occur in the ER [29,39]. 

A report from Jonas M.C. et al. [29], indicates that APP is a substrate of the ER-based acetylation machinery; this finding prompted us to verify this possibility. We immunoprecipitated APP from total cell lysates of GT1 cells treated (+) or not (−) with NSC48478, and analysed whether APP could be detected with an antibody against acetylated lysine. In agreement with the above-reported findings [29], only the immature forms of APP were found acetylated (Figure 2, arrowheads) and the upper bands were not acetylated, which correspond to mature mAPP. 

Moreover, since protein sulfation reaction occurs in the late Golgi compartments from which APP is excluded under inhibitor treatment (see next experiments), we ruled out the possibility that the upper APP bands (#) derived from APP sulfation.

On the other hand, as shown in Appendix A, we found that digestion of immunoprecipitated APP by alkaline phosphatase enzyme to cleave phosphate groups from proteins, produced a shift in APP migration on SDS-PAGE gel (compare IP lane –, black arrow with lane +, white arrowheads) indicating the presence of phosphorylated APP in GT1 cells. However, under NSC48478 (Appendix A, right panel) APP was not phosphorylated because no shift in APP migration was revealed after alkaline phosphatase digestion.

Altogether, these findings indicate that NSC48478 prevents *N*- and *O*-glycosylation of APP, as well as phosphorylation, presumably affecting its intracellular localization.

### 2.2. NSC48478 Inhibitor Induces Partial ER Retention of APP and Localization in Tfr-Positive Recycling Endosomal Structures

Next, we analysed the effect of NSC48478 on the intracellular localization of APP by fluorescence microscopy. We observed that the steady state localization of APP in control conditions was mainly restricted to endolysosomes and Golgi apparatus in GT1 cells (Figure 3).

In contrast, treatment with NSC48478 revealed a dramatic change in the intracellular localization of APP in puncta scattered in the cytoplasm and in a pattern that partially resembled the typical reticular ER structures (Figure 4A). However, biotinylation assays of proteins on the plasma membrane indicate that the presence of APP on the cell surface was not prevented by NSC48478 (Figure 4B). 

The partial ER localization was confirmed by colocalization analysis of APP with the ER resident protein KDEL receptor (Figure 4, panel NSC48478). The Pearson’s Correlation Coefficient (PCC) produced a value of 0.68, which reflects the good degree of colocalization between the two proteins. This finding confirmed that NSC48478 partially accumulates APP at the ER, and explained the resulting distinct localization of immature APP respect to control conditions. 

Moreover, the lack of overlapping between the APP puncta and KDEL (PCC close to zero) suggested that these structures are not part of the ER.

In order to characterize these structures, we employed fluorescence microscopy using different markers of intracellular organelles, finding, after NSC48478 incubation of GT1 cells, a significant colocalization of APP with the marker of the endosomal recycling compartment Tfr (Figure 5, PCC 0.89), with a concomitant loss of APP localization in the Golgi apparatus (Figure 5, compare control *versus* NSC48478, APP/Giantin).

These results strongly suggest that NSC48478 triggers the entry of APP in the ER-associated structures that favour the sorting of APP in the endosomal recycling-dependent pathway against the physiological ER to Golgi secretory pathway [4], thus impeding regular APP maturation. 

### 2.3. Effects of the Inhibitor on Both APP Maturation and Intracellular Localization Are Rescued by Controlling Endolysosomal Activity

Previous findings reported by Haass et al., [40], where it was described that sorting of APP to the plasma membrane occurs via a pH-sensitive compartment, prompted us to analyse the consequences of inhibition of vesicular acidification by NH_4_Cl when NSC48478 was administrated to GT1 cells.

In agreement with our finding of APP in the Golgi apparatus and endolysosomes (Figure 3 and Figure 5) and previous findings describing APP transport from the Golgi to lysosomes for processing and degradation [4], we found that NH_4_Cl treatment increased APP level respect to basal conditions (Figure 6). Under NSC48478 treatment, maturation of APP was completely rescued by NH_4_Cl without perturbing total APP and tubulin levels (Figure 6, bottom panel).

These results were corroborated by fluorescence microscopy (Figure 7 and Figure 8). Here, we show that under NSC48478 treatment, the partial ER localization of APP (Figure 7, compare upper and bottom panels) was completely rescued by NH_4_Cl.

Similarly, the same experiment performed by analysing the fluorescence signal deriving from APP antibody hybridization and LysoTracker (marker of endolysosomes), revealed that NH_4_Cl rescued the endolysosomal localization of APP which was lost under NSC48478 treatment (Figure 8).

### 2.4. APP and 37/67kDa LR Interact and Their Interaction Is Affected by NSC48478 

In recent years, it has been demonstrated that 37/67 kDa LR and APP share a common cellular localization and binding partners and the direct or indirect interaction between these proteins was considered possible [22,23,41]. 

In order to elucidate the relationship between APP and 37/67 kDa LR in the context of AD pathology, we analysed the structural interaction between these proteins in vitro and in neuronal GT1 cells.

To assess whether APP directly interacts with 37/67 kDa LR, the binding of human recombinant soluble 37LRP (r37LRP) to APP was evaluated in vitro by ELISA assays on cell lysates from two cell lines, the neuronal GT1 and the HeLa cell line, respectively expressing high and low APP protein levels. Purified His-tagged r37LRP and pTrc-His B, as a control, were incubated on wells pre-coated with cell lysates and binding was detected by anti-His HRP (Horseradish peroxidase). As control for binding specificity, r37LRP binding to BSA-coated wells was also evaluated in parallel and the absorbance readings subtracted. As shown in Figure 9A, r37LRP binding to cell lysate-coated wells was significantly higher in the cell lysate from neuronal GT1 cell line which express high levels of APP, compared to the cell lysate from HeLa cells expressing low levels of APP. 

These initial data provide the first evidence that APP could interact with 37/67 kDa LR. 

The identification of APP protein binding partners and the identification of small molecules capable to interfere with these interactions are an important approach to study AD. Therefore, we decided to study whether NSC48478 influenced the interaction between APP and 37/67 kDa LR.

The ability of NSC48478 to inhibit the binding of r37LRP to APP was first evaluated by ELISA assays (Figure 9B). Purified His-tagged r37LRP and pTrc-His B were incubated on wells pre-coated with GT1 cell lysate and binding was detected by anti-His HRP. r37LRP binding to BSA-coated wells was evaluated in parallel, as control, and the absorbance readings subtracted. 

r37LRP binding to the cell lysate was significantly higher than pTrc-His B and specifically inhibited by both anti-APP antibody, as control for binding specificity, and by NSC48478. 

Subsequently, to confirm the data obtained from ELISA, we performed a pull-down assay with r37LRP coupled to nickel-NTA agarose on GT1 cell lysates in both control condition and after addition of 20 μM NSC48478. As shown in Figure 9C, no APP was detectable using pTrc-His B bound agarose whereas r37LRP specifically and directly interacted with APP and this interaction could be strongly inhibited by NSC48478. Thus, NSC48478 is a specific inhibitor of 37/67 kDa LR binding to APP.

### 2.5. Effects of NSC48478 Inhibitor Are Dependent on 37/67 kDa LR Expression

To evaluate the role of the 37/67 kDa LR in the regulation of APP maturation and intracellular localization, 37/67 kDa LR was downregulated by employing short hairpin RNAs (shRNAs) (see material for description) (Figure 10). 

When compared to the shRNA scrambled (shRNA GFP), 37/67 kDa LR downregulation (by shRNA 37/67 kDa LR, 2170) hampered the inhibitor effects on APP subcellular localization (Figure 10). Indeed, the partial ER accumulation evidenced by colocalization between APP and KDEL in shRNA GFP cells treated with NSC48478 (Figure 10B, panels a,b, PCC 0.65), was completely absent in cells interfered for 37/67 kDa LR (shRNA 37/67 kDa LR)(Figure 10B, panels c,d, PCC close to zero). These results clearly indicate that the inhibitor effects are mediated by the presence of the receptor in GT1 cells.

### 2.6. Inhibitor Treatment Inactivates the MAPK-ERK1/2 Axis and Activates Akt with Consequent Inactivation of GSK3β Pathway in Neuronal Cells

It has been previously shown that a link between APP and activation of Aβ-dependent Ras-MAPK signalling pathway exists [42]. Indeed, Ras-MAPK activation is able to induce APP and tau hyperphosphorylation, which are enriched in Alzheimer’s disease brains [19]. 

In agreement with results in rat neuroblastoma B103 cells [43], where ERK1/2 was active in basic cellular metabolism, we found that GT1 cells show activation of ERK1/2, indicative of activation of Ras-MAPK signalling axis under control steady-state conditions (Figure 11A).

We found that NSC48478 administration induced the inactivation of the ERK1/2 signalling axis in a time-dependent manner (Figure 11A), whilst the total ERK1/2 levels were not affected.

In the same conditions, the analysis of Akt activation by Western blot under NSC48478 treatment, revealed that pAkt was increased respect to control conditions (Figure 11B), with concomitant 60% increase of phosphorylation in Serine 9 of GSK3β (Figure 11C), indicating that the inhibitor is acting through two signalling pathways (ERK1/2 and Akt) towards their inactivation. As expected the presence of LiCl (10 mM), which is known to inhibit the GSK pathway [44], induced an increase in phospho-Ser9 GSK3β levels without perturbing total GSK levels (Figure 11C). This effect was amplified by the contemporary presence of NSC48478 in the cell culture media, indicating that the two compounds can exert an additive effect on GSK pathway inhibition.

## 3. Discussion

The finding that the 37/67 kDa LR may play a key role in Alzheimer’s disease [22,23,26] and that it could act as a receptor mediating Aβ cytotoxicity [24,25], prompted us to verify the effects of a specific 37/67 kDa LR inhibitor on the expression levels and posttranslational modifications of APP, which are known to have a critical role in Aβ generation [14,18]. In addition, the correct localization and trafficking of proteins are fundamental for their correct processing and function [17,45].

Firstly, after verifying the canonical localization in the Golgi, as well as endolysosomal and plasma membrane distribution of APP in GT1 cells, we tested the effects of NSC48478 on APP levels, posttranslational modifications and intracellular localization. Interestingly, we found that *N*- and *O*-glycosylations of APP, as well as phosphorylation, were both inhibited by the NSC48478 compound and that these effects were due to a different intracellular localization of APP in GT1 cells. Moreover, the inhibitor seemed to be specifically affecting APP and not (or only with a milder effect) other glycoproteins, such as PrP or PLAP. Concurrently, the inhibitor was inactive in another neuronal cell type, such as neuroblastoma SHSY5Y cell line. Although we can speculate that the effects of the compound are cell type specific, this evidence deserves further investigation. 

In GT1 cells, NSC48478 prevents APP from being transported from the ER to the Golgi complex, in fact under inhibitor treatment APP does not colocalizes with Giantin, a typical Golgi marker, but partially localizes with both the ER and the recycling compartment (Figure 5). Interestingly, the recycling compartment has been already described to have a critical role in the regulation of peptides metabolism and protein function [46]. 

Since we found that APP interacts with 37/67 kDa LR and that their interaction was significantly reduced by the inhibitor, we decided to check whether NSC48478 action was mediated by 37/67 kDa LR. Interestingly, we found that the inhibitor effects were hampered by 37/67 kDa LR knockdown and that the inhibitor effects were exerted through a pH-dependent compartment [40].

Interesting studies from Hoefgen S et al., [47], report that APP shows a pH-dependent conformational switch in its N-terminal E1 domain (growth factor-like binding domain), which is able to influence its trafficking and processing. Specifically, at neutral pH 7.4 (representing the cell surface) APP may adopt a more open overall conformation than at low pH 5.7 (representing endosomal pH) [47]. 

From this evidence and our data herein shown, we can envisage the following *scenario*: (1) APP and 37/67 kDa LR are able to interact, (2) their interaction is affected by the inhibitor NSC48478 (see ELISA and pull-down assays) which diverts the “regular” APP secretory pathway (ER → Golgi → endolysosomes → plasma membrane) towards a “non-canonical” pathway (ER → recycling → plasma membrane), (3) inhibition of vesicular acidification by NH_4_Cl prevents the receptor inhibitor from acting on APP localization and maturation. This latter finding suggests that NSC48478 acts by a mechanism that, according to previous report [40], involves a pH-sensitive compartment through which APP traffics and where, based upon specific pH value, APP becomes more or less able to interact with other molecular partners because of its specific conformation. 

The IC_50_ of 19.35 mM for inhibition of receptor binding to laminin-1 raises the question of dose needed to achieve a therapeutic concentration *in vivo*. However, in disease conditions where APP is processed along a well-described pathological amyloidogenic pathway, the compound could be likely challenged with a different environment where it could affect the processing and maturation of APP hopefully requiring a lower dose to achieve any effects, for instance on Aβ generation. Moreover, among a series of previously proposed drugs against misfolding diseases, NSC48478 is a very small molecule and thus it could have the advantage to be small enough to cross the blood-brain barrier (BBB), if tested *in vivo*. It can be considered a promising compound to be tested in disease conditions where brain permeability is one of the main obstacles for molecule targeting. 

NSC48478 is essentially intended to act on CNS, so the BBB must be crossed for its effect to be executed. Accordingly, all the physicochemical parameters affecting the BBB were selected and calculated by using the QikProp software (QikProp, Schrödinger, LLC, New York, NY, USA, 2019). For small molecules in particular, lipophilicity, as measured by log *P*, can be an excellent indicator of BBB permeability. To cross the hydrophobic phospholipid bilayer of a cell membrane by passive diffusion, a molecule must be lipophilic, and given that log *P* values range for most drugs between −0.05 and 6.0 [48], the ideal range for BBB permeability has been found to be 1.5–2.5. The QP log *P* calculated for NSC48478 was 4.36, reflecting the highly lipophilic nature of the compound. Literature survey suggests that Polar Surface Area (PSA) is a measure of a molecule’s hydrogen bonding capacity and its value should not exceed certain limit if the compound is intended to be active in the CNS [49]. The most active CNS drugs have PSA of less than 70 Å^2^. The value of PSA for NSC48478 was 32.42 Å^2^, indicating good penetration through the BBB.

The log BB (the blood/brain partitioning coefficient) is the other principal descriptor to be identified as important for CNS penetration. On the basis of permeability, Vilar et al. [50] classified compounds into three categories: (a) compounds with log BB ≥ 0.3, that readily cross the BBB, (b) compounds with log BB between 0.3 and −1, which still have access to the CNS, and (c) compounds with log BB < −1, which have poor distribution in the brain. The QP log BB value for our NSC48478 was 0.008, indicating that it could easily penetrate through BBB. Madin–Darby canine kidney (MDCK) cells are considered to be a good mimic for the blood-brain barrier (for non-active transport) [51]. The higher the value of MDCK cell, the higher is the cell permeability. MDCK value of NSC48478 was 4234 nm/s (range < 25 poor; >500 great), showing excellent BBB penetration. Thus, NSC48478 possesses favorable pharmacokinetic properties, including brain penetration, which can be even further optimized in future studies. 

In order to analyse the molecular signalling connected to inhibitor treatment, and starting from previous observation of ERK1/2 pathway activation in AD patients with mild to severe pathology, together with previous reports showing 37/67 kDa LR-dependent regulation of MAPK phosphatases activity, which controls ERK signalling [52], we analysed pERK levels in inhibitor-treated GT1 cells compared to untreated control cells. Inactivation of ERK1/2 pathway in NSC48478 treated-cells was accompanied by increased Akt phosphorylation and consequent phospho-Ser9-GSK3β production (Figure 11). Since the presence in the cells of this latter isoform of GSK is indicative of Akt-GSK pathway inactivation [53], from our results showing decrement of APP phosphorylation in inhibitor-treated cells, we can conclude that NSC48478 inhibitor is able to negatively regulate GSK signalling with reduction of phosphorylated APP isoform.

The ability of NSC48478 to divert the normal traffic of APP towards an alternative pathway (ER → endosomal recycling compartment → cell surface) and the consequent inactivation of ERK and GSK pathway, can be critical for APP processing and reveal NSC48478 a useful compound to be tested in disease conditions for switching APP processing, possibly from amyloidogenic to non-amyloidogenic, and/or against Aβ production. 

Future studies will be needed to validate 37/67 kDa laminin receptor as a potential target for Alzheimer’s disease. Starting from the observation that the receptor has been previously described to be involved in Aβ generation and toxicity [23,24,25], one possibility is to challenge NSC48478 with Alzheimer’s patient cells, where Aβ levels are higher compared to that of healthy cells, and check the overall APP processing and Aβ production.

Previously proposed drugs against misfolding diseases range from small organic compounds to antibodies [54]; various therapeutic strategies have been proposed, including blocking the conversion of normal to misfolded protein, increasing clearance of amyloid aggregates, and/or stabilizing amyloid fibrils. While several compounds have been effective *in vitro* and in animal models, none have been proven effective in clinical studies to date mostly because, for many of them, it has not been discovered or described both the mechanism of action and the eventual molecular target. Such lack of in vivo efficacy is attributable to high compound toxicity and the lack of permeability of the selected compounds across the blood-brain barrier. NSC48478 (which is a naphtol derivative) has the advantages to be small enough to likely cross the blood brain barrier and to act on an already known molecular target: the 37/67 kDa LR. As a rule, the amyloid formations can be considered a target in clinical trials, however they have the disadvantage to form at advanced state of a neurodegenerative disorder and mainly consist of aggregated misfolded proteins that rarely can be effectively cleared by drug treatment.

In the case of NSC48478, we can assume that the mechanism of action can be likely oriented towards a regulation of trafficking and maturation of a specific protein (in this case APP) contrasting the association of the receptor with APP, with consequences for intracellular signaling and disease progression; thus the advantage of using this compound is that it could exert its effect at steps before the amyloid formation occurs in diseased cells.

## 4. Materials and Methods

### 4.1. Reagents and Antibodies

Cell culture reagents were purchased from Gibco Laboratories (Grand Island, NY, USA). The SAF32 anti-PrP antibody was from Cayman Chemical (Ann Arbor, MI, USA). The recombinant His-tagged 37LRP polypeptide (r37LRP) was made in bacteria and Nickel affinity purified, as previously described [27]. The polyclonal 4290 Ab was made against a C-terminal peptide derived from LR and was a kind gift from Dr. Mark E. Sobel (Bethesda, MD, USA); NSC48478 inhibitor has been already described [27]. Protein-A-Sepharose was from Pharmacia Diagnostics AB (Uppsala, Sweden). Transferrin Alexa-594- conjugated (Tfr), Alexa-488-, Alexa-546-conjugated secondary Abs and LysoTracker Red DND-99 were from Invitrogen (Molecular Probes). The anti- KDEL, anti-Giantin, antibodies were from StressGen Biotechnologies Corp (Victoria, BC, Canada). Anti-β-tubulin antibody was from Abcam. Anti-phospho RPS6 antibody (Ser240/244), anti-GSK and anti-pGSK3β Ser9 were from Cell Signalling Technology. DAPI was purchased from Cell Signal Technology. Biotin-LC was from Pierce and all other reagents were from Sigma Chemical Co. (St Louis, MO, USA). NSC48478 [1-((4-chloroanilino)methyl)-2-naphtol] was obtained from the NCI/DTP Open Chemical repository (http://dtp.cancer.gov), dissolved in DMSO and stored at −20 °C.

### 4.2. Cell Culture and Drug Treatment

GT1 (mouse hypothalamic neuronal cell line) and HeLa were grown in Dulbecco’s Modified Eagle’s Medium (DMEM), with 4500 mg/glucose/L, 110 mg sodium pyruvate and L-glutamine (Sigma-Aldrich, St. Louis, MO, USA, code D6429), supplemented with 10% fetal bovine serum. For inhibitor NSC48478 treatment, the cells were washed in serum-free medium, incubated for 30 min at room temperature in Areal medium (13.5 g/L of Dulbecco’s Modified Eagle’s Medium with glutamine SIGMA-D-7777 without NaHCO_3_, 0.2% BSA and 20 mM HEPES, final pH 7.5) and for further indicated times at 37 °C under 5% CO_2_ in the presence of 20 μM inhibitor in DMEM supplemented with 1% serum. NH_4_Cl (20 mM in culture medium) was used for 24 h. 

### 4.3. shRNA Interfering

#### Short Hairpin RNA Sequence Used

“TGCTGTTGACAGTGAGCGAGCATTGCAGAAGTGAAGGATTTAGTGAAGCCACAGATGTAAATCCTTCACTTCTGCAATGCCTGCCTACTGCCTCGGA” against mouse 37/67 kDa LR (m 2170) (from Open Biosystems Library, Euroclone, Milano, Italy) was cloned in pSHAG-MAGIC vector. GT1 cells at 65% confluence were transfected in Optimem with shRNA by Lipofectamine LTX (Invitrogen, Waltham, MA, USA, see manufacturer’s protocol). Puromycin (Sigma-Aldrich, code p8833, 2 μg/mL for 5 days) was used to select cells with integrated plasmid.

### 4.4. Deglycosylation Assays 

PNGaseF, Endo-H, and *O*-glycosidase/Neuraminidase digestions were performed as follows: for PNGaseF treatment, protein extracts were denatured at 100 °C in denaturing buffer for 10 min (as indicated in manufacturer instruction, code 1365169, Sigma-Aldrich), then treated with the enzyme in Glyco buffer with NP40 for 1 h at 37 °C. For Endo-H digestion (code 1088726, Roche, Basel, Switzerland), samples were first denatured for 3 min at 100 °C in 0.1 M sodium citrate, 0.1% SDS, then incubated with the enzyme 16 h at 37 °C. For *O*-glycosidase/Neuraminidase treatment, samples were denatured as reported in manufacturer instruction (code P0733S/P0720S New England BioLabs, Beverly, MA, USA) and digested with the enzymes 1 h at 37 °C. All samples were analysed by SDS-PAGE and Western blotting. 

### 4.5. Alkaline Phosphatase Assay

GT1 cells grown on 100 mm dishes were washed 3 times in cold PBS and lysed on ice in 1% NP-40 lysis buffer (50 mM Tri-HCl pH8, 150 mM NaCl, 1% NP-40, *w*/*o* EGTA or EDTA) with protease inhibitor mixture. APP was immunoprecipitated from 1.5 mg of total proteins, while 50 μg of total proteins were kept for reference. To immunoprecipitate APP, cell lysates were precleared with protein-A sepharose (5 mg/sample) for 30 min at 4 °C and incubated overnight with anti-APP A8717 antibody (1:300) preventively linked to protein A-Sepharose, or anti phospho-RPS6 (1:50) antibody to immunoprecipitate phosphorylated RPS6 isoform as control of the procedure. Pellets were washed 3 times with cold 1% NP-40 buffer and treated for phosphatase assay.

Reaction mixture: 1X FastAP reaction buffer, 0.2 mg/mL of protein sample, 10 U of FastAP thermosensitive Alkaline phosphatase (ThermoFisher Scientific, Waltham, MA, USA, code #EF0651). Samples were incubated 2 h at room temperature and the reaction was stopped by adding 10 mM Na_3_VO_4_. After mixing with loading buffer, samples were loaded on 8% polyacrylamide gel, transferred on PVDF. APP and pRPS6 were revealed by Western blotting with specific antibody.

### 4.6. Indirect Immunofluorescence and Confocal Microscopy

GT1 cells were cultured to 50–70% confluence in growth medium for three days on coverslips, washed in PBS, fixed in 4% paraformaldehyde (PFA), permeabilized or not with 0.1% TX-100 for 30 min (where indicated) and processed for indirect immunofluorescence using specific antibodies 30 min in PBS/BSA 0.1%. The cells were incubated with rabbit anti-APP (A8718) antibody and markers of intracellular organelles, followed by incubation with fluorophore-conjugated secondary antibodies. For lysosomal staining, cells were incubated for 1 h with LysoTracker (1:10,000) in complete medium before fixing. For Tfr-Alexa 594 staining, the cells were incubated 30 min in complete medium before proceeding with immunofluorescence. Nuclei were stained by using DAPI (1:1000) in PBS. 

Pearson’s correlation coefficient (PCC) was employed to quantify colocalization [55] between APP and KDEL (as well as other intracellular markers), and was determined in at least 25 cells from four different experiments. PCC was calculated in regions of APP and reference protein co-presence [55]. In brief, the Otsu algorithm was applied to segment APP and KDEL (as well as other intracellular markers) images, in order to define co-localization regions of the reference proteins. The PCC was then calculated in the defined regions for the images of interest. Immunofluorescences were analysed by the confocal microscope LSM 700 Zeiss equipped with an oil immersion 63 × 1.4 NA Plan Apochromat objective, and a pinhole size of one airy unit. 

Measurements of fluorescence intensity were taken on a minimum of three confocal stacks per condition, from a single experiment (∼50 cells), using LSM 510 Zeiss software. The background values raised by fluorescent secondary antibodies alone, were subtracted from all samples. 

### 4.7. LRP-His-Tag Protein and 37LRP Conjugated Agarose Beads Preparation

To investigate the interaction of APP with 37/67 kDa LR, a 37LRP-His-tag fusion protein was generated. To this end, wild-type 37LRP cDNA [56] was cloned into the pTrc-His B expression vector (Invitrogen, San Diego, CA, USA) and expressed in TOP-10 bacteria (Invitrogen) and the resulting plasmid was named pPLR2-1. According to the procedures specified by Invitrogen, TOP-10 bacteria were transformed with pPLR2-1 and pTrc-His B alone, as a control, and lysed in a denaturing lysis buffer (20 mM sodium phosphate, 500 mM sodium chloride, pH 7.8) containing 6M guanidium. Both bacterial lysates were bound to nickel-NTA agarose beads through their His-tagged N-terminus in the same denaturing buffer containing 8M urea. Beads were washed several times at pH 6.0 and 5.3, to dissociate contaminating proteins and His tagged proteins were eluted at pH 4.0. His tagged recombinant 37LRP (r37LRP) purity was than 90% pure, as assessed by SDS-PAGE and Coomassie stain, as compared to pTrc-His B eluate. 

r37LRP conjugated beads and pTrc-His B-conjugated control beads, produced as described above, were washed in 50 mmol/L Tris (pH 7.5)-0.1% Triton X-100, to remove urea, and resuspended in the same buffer for pull-down assays.

### 4.8. Binding of Soluble r37LRP to Immobilized GT1 and HeLa Cell Lysates

High binding plates with 96 flat-bottomed wells (Corning, Amsterdam, Netherlands) were coated with GT1 and HeLa cell lysates, or BSA as a negative control, and incubated at 4 °C overnight. After a wash in PBS, residual binding sites were blocked for 1 h at 37 °C with 200 μL of blocking buffer (2% FCS, 1 mg/mL BSA, in PBS). Wells were incubated with 2 μg of pTrc-His B control eluate (diluted in PBS, 1 mg/mL BSA), or 2 μg of r37LRP (diluted in PBS, 1 mg/mL BSA), which both contained a 6 × His-tag, for 1 h at 37 °C. Each well was washed three times with wash buffer (0.5% Tween in PBS). Penta-His HRP conjugate (1:500) (Qiagen, Hilden, Germany) was added for 2 h at room temperature. After washing, substrate solution was added and absorbance was detected at 490 nm on an ELISA plate reader (Bio-Rad, Hercules, CA, USA). Binding affinity was determined by subtracting background absorbance (BSA wells). 

For inhibition experiments, wells pre-coated with GT1 cell lysates were incubated with 2 μg of pTrc-His B, as control, and 2 μg of r37LRP, alone and in the presence of APP antibody (1:1000), or NSC48478 compound (20 μM).

### 4.9. Pull-Down Assay

Neuronal GT1 cell line was harvested into 500 μL of magnesium lysis buffer (MLB, 125 mM HEPES, pH 7.5, 750 mM NaCl, 5% Igepal CA-630, 50 mM MgCl_2_, 5 mM EDTA and 10% glycerol), containing protease and phosphatase inhibitors. 

Total cell lysates were pre-cleared with 50 μL of nickel-NTA agarose beads (Invitrogen) overnight at 4 °C.

After precleaning, cell lysates were incubated in the presence of 50 μL of agarose-bound r37LRP (approximately 2 μg) or in the presence of 50 μL of agarose-bound His pTrc-His B, as a control, for 2 h at 4 °C. Beads were washed five times with MLB, and then resuspended in Laemmli buffer (4% SDS, 20% glycerol, 10% 2-mercaptoethanol, 0.004% bromphenol blue and 0.125 M Tris HCl) followed by boiling for 5 min and centrifugation at 25,000× *g* for 3 min. Supernatants were analysed by SDS-PAGE and blotted with anti-APP antibody. Separately, 50 μg of total cell lysate were immunoblotted for APP.

### 4.10. Biotinylation Assay

GT1 cells grown on dishes were cooled on ice and biotinylated with NHS-LC-Biotin at 4 °C (as previously indicated [57,58]. Cells were lysed for 20 min using buffer 1 (25 mM Tris-HCl pH 7.5), 150 mM NaCl, 5 mM EDTA, 1% TX-100). Biotinylated cell surface proteins were immunoprecipitated with streptavidin beads (40 μL/sample, Pierce, ThermoFisher Scientific, code n. 20349). APP was specifically immunorevealed with the A8717 antibody. In the case of NSC48478 treatment, the cells were first incubated with the inhibitor (or not, control) for 24 h, then biotinylated on ice, following the protocol described above. 

### 4.11. Statistical Analysis 

Statistical significance of samples against untreated cells was determined by one-way analysis of variance (ANOVA), followed by the Dunnett test. Each value represents the mean ± SEM of at least three independent experiments performed in triplicate (* *p* < 0.05). 

## Figures and Tables

**Figure 1 ijms-21-01738-f001:**
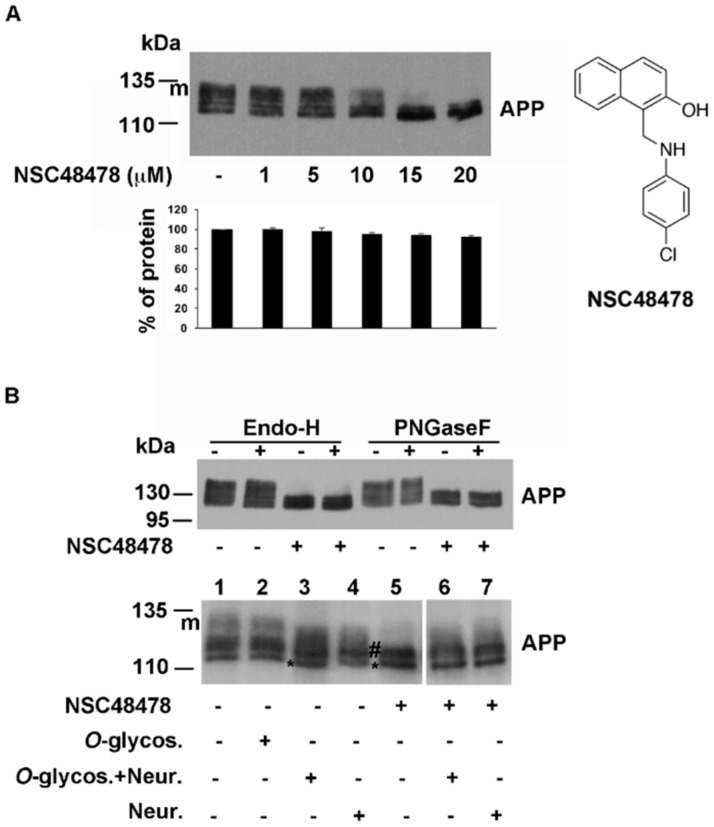
NSC48478 inhibitor affects maturation of APP in neuronal cells. (**A**) GT1 cells grown in DMEM (Dulbecco’s modified Eagle medium) supplemented with 10% foetal bovine serum were scraped in lysis buffer and 40 μg of total proteins were subjected to SDS-PAGE. APP was revealed by Western blotting on PVDF (Polyvinylidene Difluoride membranes) and hybridization with A8717 antibody. NSC48478 was used at different indicated concentrations for 24 h. Protein levels of APP were calculated by densitometric analysis with ImageJ software and expressed as percentage. The plot shows the percentage of APP after indicated concentration of NSC48478 treatment, using as 100% the expression value in control conditions. Data are expressed as the means ± SEM of three independent experiments (**p* < 0.05). (**B**) GT1 cells were either untreated or digested for 16 h with PGNaseF, EndoH, Neuraminidase and/or *O*-glycosidase. After lysis in buffer 1, cell extracts were incubated with the specific buffers (see materials and methods for details) and APP was revealed by SDS-PAGE and Western blotting. Asterisk (*) points to immature unglycosylated APP; # indicates intermediate, not *O*-glycosylated APP isoform; “m” points to mature glycosylated APP. Plus (+) and minus (-) indicate the presence or absence of NSC48478 and enzymes. Structure of NSC48478 is reported on the right.

**Figure 2 ijms-21-01738-f002:**
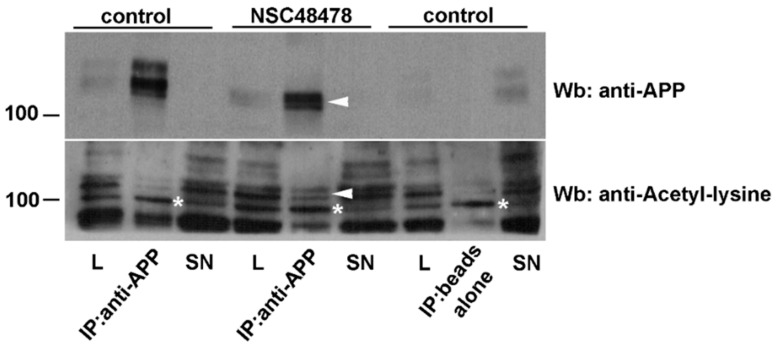
The immature APP isoforms accumulating under NSC48478 are acetylated. Total cell lysates were immunoprecipitated with an anti-APP antibody and then analysed for both APP (upper panel) and anti-acetylated lysine (lower panel) antibodies. Only the immature APP isoforms are acetylated (arrowheads). Beads alone indicate the negative control of the IP (immunoprecipitate). Asterisks (around 100 kDa) point to a specific band deriving from anti-acetylated lysine antibody hybridization. L: cell lysate; SN: supernatant.

**Figure 3 ijms-21-01738-f003:**
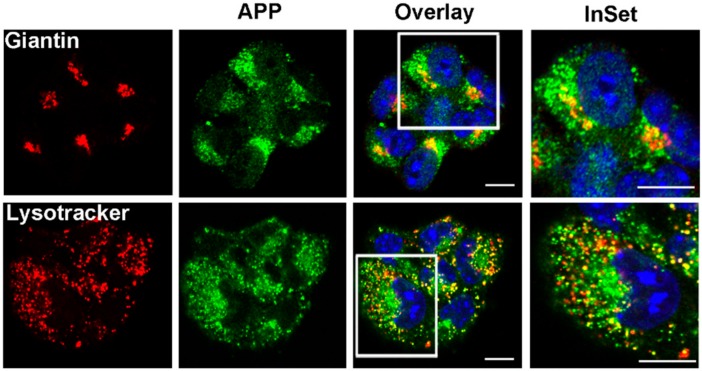
APP is localized in the Golgi apparatus and in the endolysosomal compartment. GT1 cells were grown on coverslips, fixed in PFA 4% and permeabilized in 0.1% TX-100 for 30 min, then they were stained with A8717 rabbit Ab (1:500) and Giantin (1:50) to label APP and Golgi, respectively. LysoTracker was used for 1 h in cell culture medium before fixation to label endolysosmes. Images are representative of at least 100 cells analysed. Scale bars, 10 μm.

**Figure 4 ijms-21-01738-f004:**
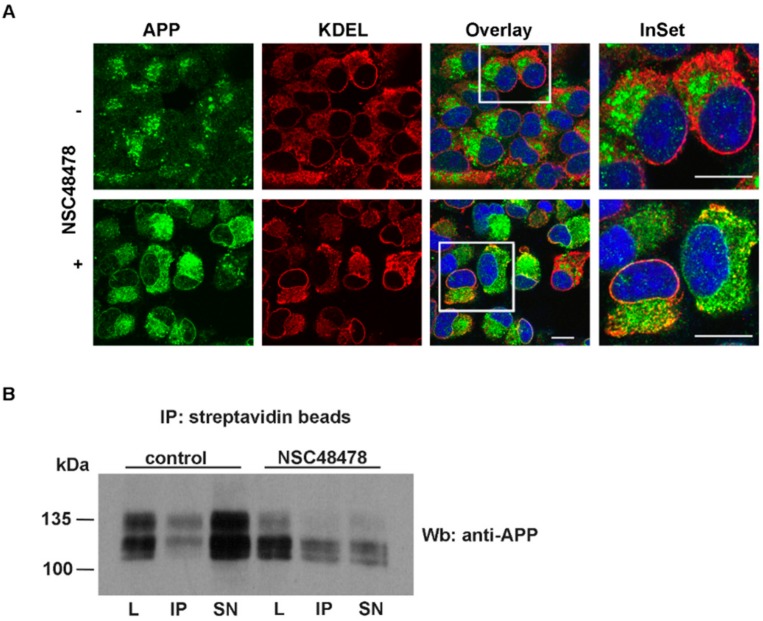
NSC48478 inhibitor induces partial ER (Endoplasmic Reticulum) retention of APP without affecting cell surface distribution of APP. (**A**) GT1 cells grown on coverslips, either untreated or treated with NSC48478 for 24 h, were subjected to immunofluorescence analysis by using anti-KDEL and anti-APP antibodies. Colocalization between APP and KDEL was then measured as indicated in the methods section. Plus (+) and minus (-) indicate the presence or absence of the compound. Scale bars, 10 μm. (**B**) Cell surface proteins were biotinylated at 4 °C in control (without inhibitor) or after treatment with NSC48478, and were recovered from cell lysates by immunoprecipitation with streptavidin-beads. Total (40 μg of total cell lysates) and cell surface proteins (IP from streptavidin beads), were loaded on gel and processed for SDS-PAGE and ECL. APP was immunodetected by blotting with A8717 Ab. L: cell lysate; IP: immunoprecipitated biotinylated proteins by streptavidin beads; SN: supernatant.

**Figure 5 ijms-21-01738-f005:**
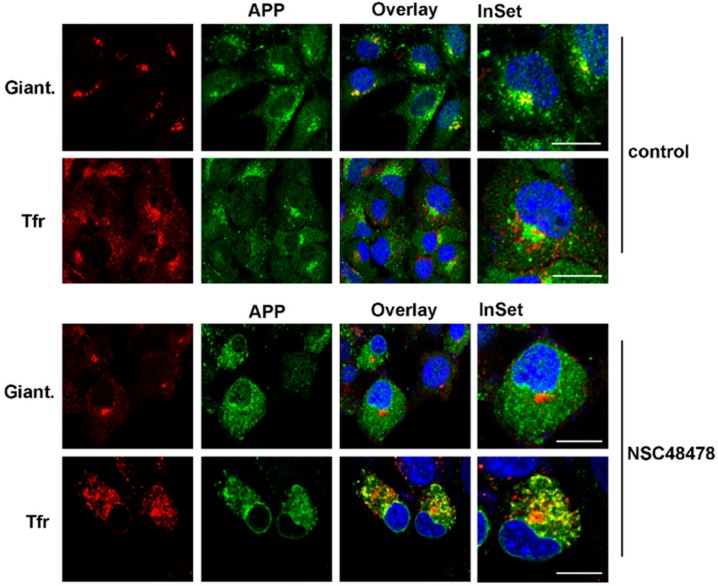
Localization of APP in the Golgi apparatus was lost, under NSC48478, in favour of Tfr-enriched endosomal compartment redistribution. GT1 cells grown on coverslips, either untreated or treated with NSC48478 for 24 h, were subjected to immunofluorescence analysis by using anti-Giantin and anti-APP antibodies. TfrAlexa-594 in the cell culture media was used to label recycling endosomes. Colocalization between APP and the different markers was then measured as indicated in the methods section. Giant: Giantin. Scale bars, 10 μm.

**Figure 6 ijms-21-01738-f006:**
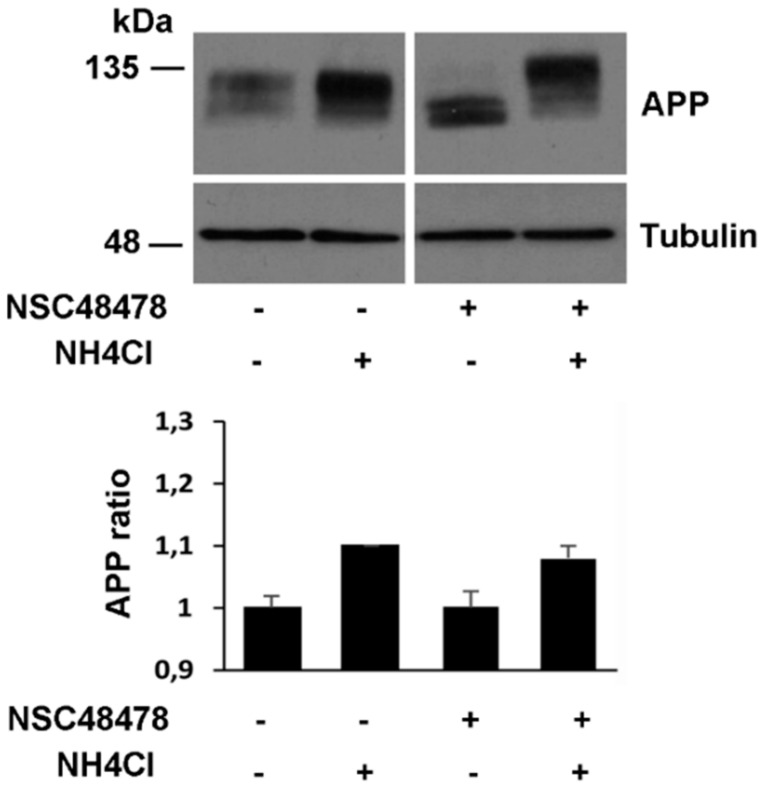
NH_4_Cl-induced acidification rescues inhibitor effects on APP maturation. GT1 cells grown on dishes, were treated or not with the inhibitor NSC48478 in the presence or absence of NH_4_Cl (see methods). The cells were scraped in lysis buffer 1 and 40 μg of total proteins were subjected to SDS-PAGE. APP and tubulin (as loading control) were revealed by Western blotting on PVDF and hybridization with A8717 and anti-tubulin Ab, respectively. Protein levels of APP were calculated by densitometric analysis with ImageJ software and expressed as a ratio, which was determined by imposing as 100% (ratio 1) the signal of APP in the untreated cells (lane 1). Mean ± SEM of three experiments were considered (*p* < 0.05). All data were statistically significant. Plus (+) and minus (-) indicate the presence or absence of NSC48478 or NH_4_Cl.

**Figure 7 ijms-21-01738-f007:**
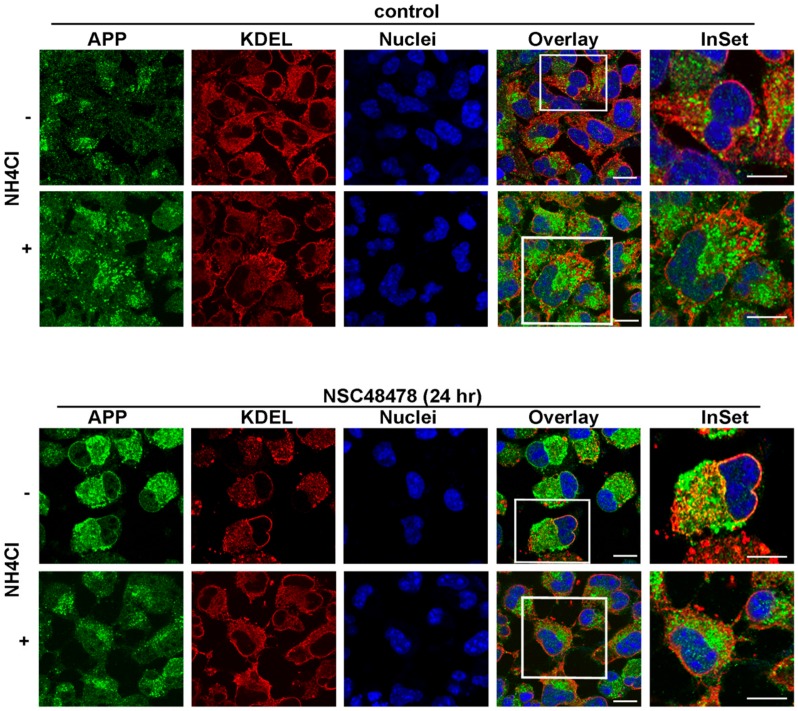
NH_4_Cl rescues inhibitor effects on APP subcellular localization. GT1 cells grown on coverslips were treated or not with the inhibitor NSC48478 in the presence or absence of NH_4_Cl. GT1 cells were subjected to immunofluorescence analysis by using anti-KDEL and anti-APP antibodies. Colocalization between APP and KDEL was then measured as indicated in the methods section. Plus +) and minus (-) indicate the presence or absence of NH_4_Cl. Scale bars, 10 μm.

**Figure 8 ijms-21-01738-f008:**
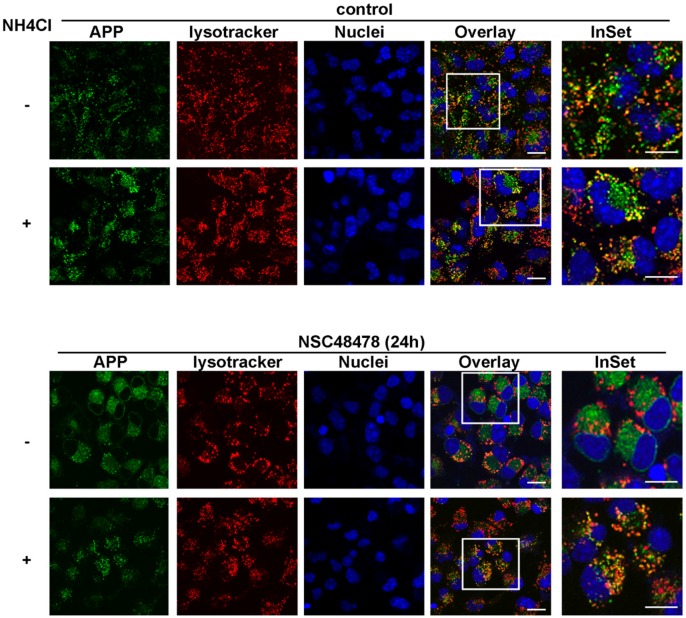
NH_4_Cl rescues APP localization in endolysosomal compartment. The cells were treated as above, with the exception that LysoTracker was added in vivo before fixation and immunofluorescence analysis. Plus (+) and minus (-) indicate the presence or absence of NH_4_Cl. Scale bars, 10 μm.

**Figure 9 ijms-21-01738-f009:**
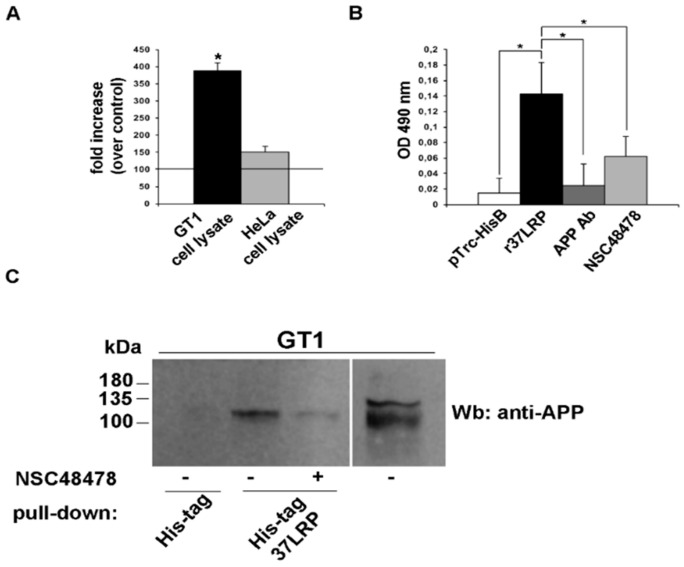
NSC48478 inhibitor affects interaction between APP and 37/67 kDa LR in neuronal cells. (**A**) Purified human His-tagged recombinant 37LRP (r37LRP) and control pTrc-His B were placed for 1 h at 37 °C on wells coated with 2 μg of GT1 or HeLa cell lysates. Bound r37LRP was revealed by anti-His-HRP and OPD staining; the absorbance at 490 nm was measured. r37LRP binding to BSA-coated wells was subtracted to obtain specific binding. Results are expressed as a percent increase of absorbance value over pTrc-His control (horizontal line represents control absorbance value). Values represent the mean ± SEM of three experiments carried out in triplicate; (* *p* < 0.05). (**B**) Purified human His-tagged recombinant 37LRP (r37LRP) and control pTrc-His B were placed for 1 hour at 37 °C on wells coated with 2 μg of GT1 cell lysates in the presence of anti-APP antibody, or NSC48478, or DMSO as a vehicle control. Bound r37LRP was revealed by anti-His-HRP and OPD staining; the absorbance at 490 nm was measured. r37LRP binding to BSA-coated wells was subtracted to obtain specific binding. Values represent the mean ± SEM of three experiments carried out in triplicate; (* *p* < 0.05). (**C**) Lysates from GT1 cells were incubated with agarose-bound recombinant His-tagged 37LRP (His-tag 37LRP) or with agarose bound His-tag (His-tag), as a negative control. His-tag 37LRP conjugated beads were washed, resuspended in Laemmli sample buffer, boiled and supernatants were analysed by 15% SDS-PAGE and blotting with anti-APP antibody. Separately, 50 μg of total GT1 lysate were immunoblotted with anti-APP antibody. Plus (+) and minus (-) indicate the presence or absence of NSC48478.

**Figure 10 ijms-21-01738-f010:**
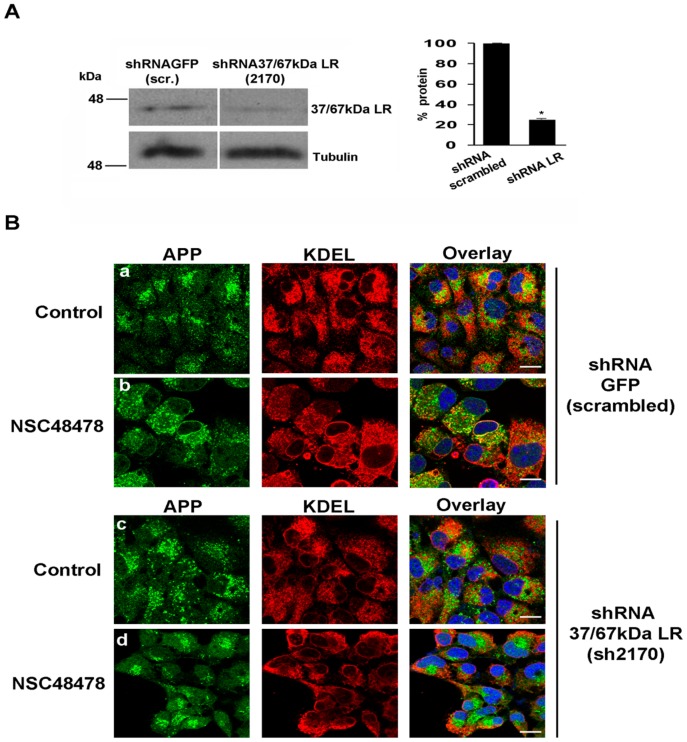
Downregulation of 37/67 kDa LR by short hairpin RNA, hampers the effects exerted by the inhibitor. (**A**) 37/67 kDa LR was silenced using a specific shRNA (mouse shRNA 2170) for 48 h. In comparison to shRNA 37/67 kDa LR (mouse shRNA 2170), 35 μg of total cell lysate from control cells (non-targeting RNA, shRNA GFP, scrambled) were loaded for reference. Membranes were probed with anti-37/67 kDa LR antibody to reveal the receptor. The same membranes were probed with anti-tubulin antibody followed by ECL. The amount of silenced receptor was quantified from three independent experiments (* *p* < 0.05). (**B**) Immunofluorescence analysis of scrambled or receptor silenced cells was conducted to analyse the effects of NSC48478 on APP localization. The cells were processed as in Figure 4. Colocalization analysis has been described in the methods section. Scale bars, 10 μm.

**Figure 11 ijms-21-01738-f011:**
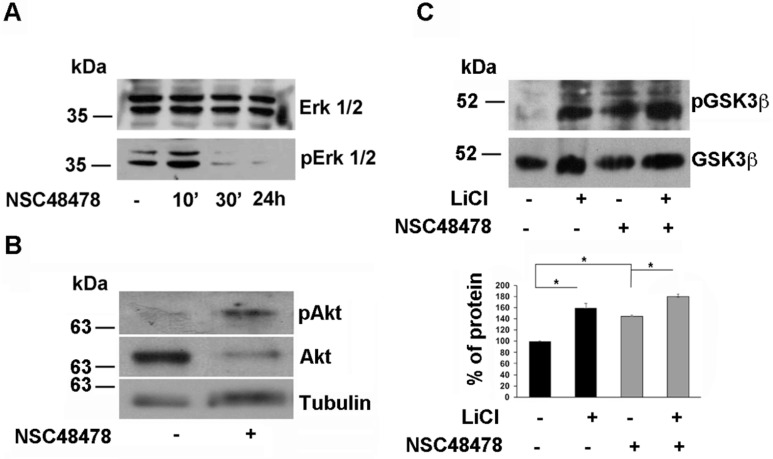
NSC48478 induces inactivation of ERK1/2 axis and activation of Akt with consequent inactivation of GSK3β. (**A**) Total cell lysates (40 μg) from cells treated or not with NSC48478 for indicated times, were loaded on gels and expression levels of both total ERK1/2 and pERK1/2 were analysed by SDS-PAGE followed by Western blotting and hybridization of PVDF membranes by respective antibodies. (**B**) Total cell lysates (40 μg) from cells treated or not with NSC48478, were loaded on gels and expression levels of both total Akt and pAkt were analysed by SDS-PAGE followed by Western blotting and hybridization of PVDF membranes by respective antibodies. Anti-tubulin antibody was used to test loading controls. (**C**) The membranes were treated as in (**B**) with the exception that here the cells were incubated with LiCl (10 mM for 24 h) to inhibit GSK3β pathway, in the presence or absence of NSC48478. Both total and pGSK3β levels were revealed by using anti- GSK3β and anti-phospho-Ser9 GSK3β antibody, respectively. The amount of pGSK3β was quantified as “percent of protein” in the plot from three independent experiments (* *p* < 0.05).

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
