# Peer review of "APP Maturation and Intracellular Localization Are Controlled by a Specific Inhibitor of 37/67 kDa Laminin-1 Receptor in Neuronal Cells"

_ijms, 2020, doi:10.3390/ijms21051738_

Round 1
Reviewer 1 Report
This manuscript describes how the maturation and intracellular localisation of amyloid precursor protein maturation is controlled by a specific inhibitor of 37/67Da laminin-1 receptor in neuronal cells. It has a number of shortcomings:
The abstract contains the abbreviation APP. This needs to be spelt out in full. The chemical name and origin of NSC48478 needs to be included. NSC48478 is described as a ‘drug’. This is not correct as it has not obtained regulatory approval. The receptor cell adhesion to laminin-1 IC50 value for NSC48478 is stated as 19,35. I assume that the comma should be a period. This IC50 value seems high and raises the question of the dose needed to achieve a therapeutic concentration in vivo. It would be helpful to include some discussion on this point and the likelihood of NSC48478 crossing the blood-brain barrier. The authors propose 37/67Da laminin-1 receptor as a molecular target for drug discovery efforts for new drugs to slow disease progression in Alzheimer’s disease. It would be helpful to include a description of future studies to validate this target. The authors should describe the advantages of inhibiting the receptor cell adhesion to laminin-1 over amyloid-targeted compounds already used in clinical trials to assess a slowing of disease progression. This is important as all of the compounds assessed failed to show any therapeutic efficacy sufficient to gain regulatory approval. Some description of such studies need to be added.
Author Response
Point-by-point response to reviewers
Reviewer# 1
This manuscript describes how the maturation and intracellular localisation of amyloid precursor protein maturation is controlled by a specific inhibitor of 37/67Da laminin-1 receptor in neuronal cells. It has a number of shortcomings:
The abstract contains the abbreviation APP. This needs to be spelt out in full.
Response: done.
The chemical name and origin of NSC48478 needs to be included. NSC48478 is described as a ‘drug’. This is not correct as it has not obtained regulatory approval.
Response: we thank the referee for this annotation. We have now provided full name of the compound NSC48478 [1-((4-chloroanilino)methyl)-2-naphtol] and its origin (pag16, line 500-502), in the methods section of the revised manuscript. The compound was obtained from the NCI/DTP Open Chemical repository (http://dtp.cancer.gov), dissolved in DMSO and stored at -20°C. We have now replaced (through the entire manuscript) the term “drug” with “compound” when refer to NSC48478.
The receptor cell adhesion to laminin-1 IC50 value for NSC48478 is stated as 19,35. I assume that the comma should be a period. This IC50 value seems high and raises the question of the dose needed to achieve a therapeutic concentration in vivo. It would be helpful to include some discussion on this point and the likelihood of NSC48478 crossing the blood-brain barrier.
Response: we apologize for the typographical error and thank the referee for noting that. We have now replaced the comma with a period in the IC50 value 19.35 mM (Pag 3, line 92).
The inhibitor exerted the maximum of effects on APP maturation at concentration of 20 mM, and we agree with the referee that this could induce to the suspect that a massive dose is necessary to obtain a therapeutic concentration in vivo. We have discussed this issue in the main text (Pag 14 line 408-432) as follows: “However, in disease conditions where APP is processed along a well-described pathological amyloidogenic pathway, the compound could be likely challenged with a different environment where it could affect the processing and maturation of APP hopefully requiring a lower dose to achieve any effects, for instance on Aß generation. Moreover, among a series of previously proposed drugs against misfolding diseases, NSC48478 is a very small molecule and thus it could have the advantage to be small enough to cross the brain blood barrier, if tested in vivo. It can be considered a promising compound to be tested in disease conditions where brain permeability is one of the main obstacle for molecule targeting.
NSC48478 is essentially intended to act on CNS, so the blood–brain barrier (BBB) must be crossed for its effect to be executed. Accordingly, all the physicochemical parameters affecting the BBB were selected and calculated by using the QikProp software (QikProp, Schrödinger, LLC, New York, NY, 2019). For small molecules in particular, lipophilicity, as measured by log P, can be an excellent indicator of BBB permeability. To cross the hydrophobic phospholipid bilayer of a cell membrane by passive diffusion, a molecule must be lipophilic, and given that log P values range for most drugs between -0.05 and 6.0 [Begley DJ., Pharmacol Ther. 2004;104(1):29–45] the ideal range for BBB permeability has been found to be 1.5–2.5. The QP log P calculated for NSC48478 was 4.36, reflecting the highly lipophilic nature of the compound. Literature survey suggests that Polar Surface Area (PSA) is a measure of a molecule’s hydrogen bonding capacity and its value should not exceed certain limit if the compound is intended to be active in the CNS [Clark D., J. Pharm. Sci., 8 (88) (1999), pp. 815-821]. The most active CNS drugs have PSA of less than 70 Å2. The value of PSA for NSC48478 was 32.42 Å2, indicating good penetration through the BBB.
The log BB (the blood/brain partitioning coefficient) is the other principle descriptor to be identified as important for CNS penetration. On the basis of permeability, Vilar et al. [J. Mol. Graphics Modell. 2010, 28, 899−903] classified compounds into three categories: (a) compounds with log BB ≥ 0.3, that readily cross the BBB, (b) compounds with log BB between 0.3 and −1, which still have access to the CNS, and (c) compounds with log BB < −1, which have poor distribution in the brain. The QP log BB value for our NSC48478 was 0.008, indicating that it could easily penetrate through BBB. Madin–Darby canine kidney (MDCK) cells are considered to be a good mimic for the blood-brain barrier (for non-active transport) [Wang Q, (2005) Int J Pharm 288:349–359]. The higher the value of MDCK cell, the higher is the cell permeability. MDCK value of NSC48478 was 4234 nm/s (range< 25 poor; > 500 great), showing excellent BBB penetration. Thus, NSC48478 possesses favorable pharmacokinetic properties, including brain penetration, which can be even further optimized in future studies.
The authors propose 37/67Da laminin-1 receptor as a molecular target for drug discovery efforts for new drugs to slow disease progression in Alzheimer’s disease. It would be helpful to include a description of future studies to validate this target.
Response: Discussion has been improved as well (pag 15, line 459-463). “Future studies will be needed to validate 37/67kDa laminin receptor as target for Alzheimer’s disease. Starting from the observation that the receptor has been previously described to be involved in Aß generation and toxicity (Jovanovic 2013, Da Costa Dias 2013, Pinnock 2016), one possibility is to challenge NSC48478 with Alzheimer’s patient cells, where Aß levels are higher compared to that of healthy cells, and check the overall APP processing and Aß production”.
The authors should describe the advantages of inhibiting the receptor cell adhesion to laminin-1 over amyloid-targeted compounds already used in clinical trials to assess a slowing of disease progression. This is important as all of the compounds assessed failed to show any therapeutic efficacy sufficient to gain regulatory approval. Some description of such studies need to be added.
Response: description of studies suggested by the referee have been added to the Discussion section (pag 15, line 464-484) as follows: “Previously proposed drugs against misfolding diseases range from small organic compounds to antibodies (Eisele YS et al., Nat Rev Drug Discovery 2015); various therapeutic strategies have been proposed, including blocking the conversion of normal to misfolded protein, increasing clearance of amyloid aggregates, and/or stabilizing amyloid fibrils. While several compounds have been effective in vitro and in animal models, none have been proven effective in clinical studies to date mostly because, for many of them, it has not been discovered or described both the mechanism of action and the eventual molecular target. Such lack of in vivo efficacy is attributable to high compound toxicity and the lack of permeability of the selected compounds across the blood-brain barrier. NSC48478 (which is a naphtol derivative) has the advantages to be small enough to likely cross the blood brain barrier and to act on an already known molecular target: the 37/67kDa LR. As a rule, the amyloid formations can be considered a target in clinical trials, however they have the disadvantage to form at advanced state of a neurodegenerative disorder and mainly consist of aggregated misfolded proteins that rarely can be effectively cleared by drug treatment.
In the case of NSC48478, we can assume that the mechanism of action can be likely oriented towards a modification of trafficking and maturation of a specific protein (in this case APP) contrasting the association of the receptor with APP with consequences for intracellular signaling and disease progression; thus the advantage of using this compound is that it could exert its effect at steps before the amyloid formation occurs in diseased cells.”
Sincerely,
Daniela Sarnataro

Reviewer 2 Report
Based on an earlier report, the authors decided to test the operation of a specific LR 37 / 67kDa
Inhibitor, NSC48478, in mouse SHSY5Y neuronal cell lines. These studies are very useful for developing a new drug for the treatment of AD. The manuscript is very well prepared without errors.
I suggest only some technical correction
-line 55, 220, 224 and in all manuscript: NH4Cl – 4 should be in the index
- line 81: (NSC48478 -) – please delete “-“
-line 89: 19,35 – it should be in English style with dot
-line 106: (untreated cells, -) 
– please delete “-“
- line 480-482 – please check interline
Author Response
Point-by-point reply to reviewers
Reviewer#2
Based on an earlier report, the authors decided to test the operation of a specific LR 37 / 67kDa
Inhibitor, NSC48478, in mouse SHSY5Y neuronal cell lines. These studies are very useful for developing a new drug for the treatment of AD. The manuscript is very well prepared without errors.
I suggest only some technical correction
-line 55, 220, 224 and in all manuscript: NH4Cl – 4 should be in the index. Response: we thank the referee for noting that. Accordingly, we have replaced NH4Cl with NH4Cl through the manuscript.
- line 81: (NSC48478 -) – please delete “-“. Response: Done.
-line 89: 19,35 – it should be in English style with dot. Response: Done.
-line 106: (untreated cells, -) 
– please delete “-“. Done.
- line 480-482 – please check interline. Response: we checked and adjusted the interline. Done.
Sincerely,
Daniela Sarnataro

Round 2
Reviewer 1 Report
The authors have addressed my concerns and the revised manuscript is now acceptable for publication